# Environmental Justice and Urban Parks. A Case Study Applied to Tarragona (Spain)

Joan Alberich *, Yolanda Pérez-Albert, José Ignacio Muro Morales and Edgar Bustamante Picón

Grup de Recerca d'Anàlisi Territorial i Estudis Turístics (GRATET), Department de Geografia, Universitat Rovira i Virgili, Vila-Seca, 43480 Tarragona, Spain; myolanda.perez@urv.cat (Y.P.-A.); joseignacio.muro@urv.cat (J.I.M.M.); edgar.bustamante@urv.cat (E.B.P.)
* Correspondence: joan.alberich@urv.cat

**Abstract:** There is consensus regarding the fact that urban green areas contribute to the quality of life of their inhabitants. Therefore, efficient city management must assess whether the population has access to green areas and the areas' quality in relation to, for example, vegetation, facilities or furnishings. Therefore, the objective is to establish environmental justice of urban parks in Tarragona (Spain) by developing a Park Quality Index (PQI) and the sociodemographic characteristics (level of studies, Human Development Index –HDI–, home sale and rental prices) of the population living within 300 m of a park. To prepare this, a GIS-integrated Multi-Criteria Evaluation (MCE) was produced. The results show that the green areas have low accessibility and availability and that most parks obtain an average-low PQI, with the best-valued aspect being the vegetation and the worst being the facilities. Regarding the degree of environmental justice, a causal relationship between the PQI and the indicators used emerges. The average value of the home sale prices is the one that shows the greatest correlation. These results can be used together with participatory procedures as a basis for identifying places with greater inequality, and for selecting the more effective actions that enable increasing environmental justice with respect to green areas.

**Keywords:** environmental justice; urban parks; ecosystem services; Tarragona



## 1. Introduction

The concept of environmental justice appeared in the last third of the 20th century, within the framework of "assessing the distribution of the benefits and damage caused by human agents between places and population groups, in order to determine whether or not serious discrimination exists" [1]. The general context was a growing awareness that the spatial distribution of some human activities was clearly discriminatory for one part of the population. For example, the generation, handling and storage of hazardous waste or the territorial distribution of certain pollutant industries tended to be located in areas occupied by the less favored part of the population. Environmental justice considers that "there is a universal right to nature" on all levels (individual, family, community, etc.), with the environment being understood as a common good [2]. Therefore, the basis of the concept is the non-discriminatory distribution of environmental benefits and damages and the need to establish participative decision mechanisms "that can distribute those benefits and damages equitably among a justice community made up of located entities (subjects and objects), both current and future, who may have unequal rights and obligations" [1].

From this perspective, environment justice or discrimination can be measured, in general terms, by calculating the overall computation (social, territorial and temporal) of the environment costs and benefits (which, in economic terms, are often called "externalities") generated by a certain activity or project, so as to later clarify whether the distribution of these elements among the various groups that may be affected by said activity in some way is fair. However, other authors uphold the inclusion of other, non-economic elements in

the analysis, and choose to use indicators and variables of a different level of measurement, by applying multi-criteria analysis [3].

Different authors claim that, traditionally, the study of environmental justice has focused on analyzing the distribution of facilities with toxic emissions, waste dumps and other environment hazards that are disproportionately close to socially disadvantaged groups [4,5]. However, recent works have extended the scope of this concept to include terms such as equitable access to green areas and other natural resources [6,7]. This new interest is related to the conviction that urban parks or urban green areas help to increase the quality of life of city inhabitants because contact with urban nature as public parks promotes well-being and human health in cities [8] and urban residents can receive daily benefits [9].

An important issue shared by the literature we consulted is the actual definition of an urban park. Even though there is no single definition, the one provided by Jennings et al. [7] is considered appropriate (as it is necessarily broad), and states that urban parks are "a kind of green area that is generally public property and, consequently, accessible to the general public; and can include children's parks, leisure facilities and other characteristics that promote open air recreation". In order to analyze these urban green areas, they have been divided into categories according to their surface area and function in the urban space, according to their contents, different services, uses and the social values that they provide for different segments of the population [10,11].

There is consensus over the fact that, broadly speaking, ecosystem services imply benefits [12] in six different areas [13]: (1) they help to fight pollution [14] and contribute to microclimate normalization [15,16]; (2) reduce noise [17]; (3) improve the population's emotional wellbeing and psychophysiological balance by increasing the feeling of security [18]; (4) improve mental and physical health [19]; (5) promote outdoor life and social meetups [20] and (6) increase citizens' environmental awareness [9,21]. Therefore, ecosystem services regulate temperature and humidity, produce oxygen and filter radiation, absorb pollutants and muffle noise and, in addition, they provide an area for walking, relaxation and leisure. However, beyond their intrinsic value (their good organization, quality level or degree of protection), it is often their symbolic dimension which makes them places citizens appreciate.

Therefore, the World Health Organization considers urban green areas to be essential due to their inherent benefits for physical and emotional wellbeing [22]. A large portion of the works published on environmental justice regarding urban parks adopt a quantitative perspective. Qualitative approaches to this question are harder to find. In this respect, a notable exception is the work by Smiley, Sharma, Steinberg, Hodges-Copple, Jacobson and Matveeva (2016) [23], who analyze the opinions and preferences of minority ethnic groups regarding the use of the urban parks in Houston (Texas) using data obtained from two ad hoc surveys. From a quantitative orientation, GIS has been used to process the information. However, some authors have criticized the use of these tools, arguing that they specify the geographical units and threshold distances inappropriately and ignore the actual movement by people. Therefore, in some recent research, people have opted to use the georeferenced data produced by mobile telephones to obtain behavior patterns within green areas. These emphasize the real activities by park users, in terms of both space and time [24]. Another option is the application of a public participation GIS, such as the one used by Laatikainen, Tenkanen, Kyttä and Toivonen (2015) [25], which can provide an alternative to obtaining multifaceted knowledge on accessibility patterns. To establish the relationship between the distribution and quality of the parks and the population's socio-demographic characteristics, the Pearson correlation coefficient [26], the index of dissimilarity or an analysis of conglomerates is used [27]. Often, these parameters are accompanied by the use of statistical indicators to measure the significance of the observed differences, such as the Gini coefficient or the analysis of variance (ANOVA) test [10,28]. Equally, in comparative works between two or more cities, logistic regression techniques have been used to control and neutralize the different characteristics of the urban fabric

among the study cases [29]. Finally, another methodological aspect refers to the actual measurement of the social and environmental quality of the urban parks. In relation to this, some authors express the need to measure six parameters: access, services, security, social inclusion, visual and aesthetic quality and, finally, the ecological function [30].

From this perspective, in order to focus on efficient city management of community interests, we have to assess whether the population has access to green areas and, in addition, the quality of these areas in terms of, inter alia, the existence of vegetation and available or existing facilities or street furnishings. Therefore, the general objective of this work is to establish the degree of environmental justice in the urban parks in the city of Tarragona by establishing a Park Quality Index (PQI) and learning about the population's socio-demographic characteristics. To do this (1) a Multi-Criteria Evaluation (MCE) model was constructed within a GIS, which allows us to establish the PQI, (2) indirect, standardized indicators were determined for the socio-economic characteristics of the population living within 300 m or less of a park, such as the Synthetic Training Index (STI), the Human Development Index (HDI) or home sales prices in each sector, and (3) the PQI was correlated with the population's socio-economic characteristics to obtain the spatial justice results in terms of the availability and quality of urban green areas. The work is organized into six sections, plus the bibliography. The introduction reflects on the concept of environmental justice and reviews the methodological and conceptual aspects; the second section introduces the area of study; the third details the methodology stages and the tools used; the fourth reveals the results; the fifth contains the discussion and the final section includes the conclusions.

## 2. Area of Study: The Green Areas and the Urban Parks in Tarragona

The city of Tarragona is part of an urban area with nearly 380,000 inhabitants, with 16 municipalities and a surface area of just over 350 square kilometers. In 2019, the municipality of Tarragona had a population of 142,859 inhabitants, who, when distributed over the 57.88 square kilometers of their municipal area, represent a density of 2.468 inhab./km$^2$. However, this average value does not reflect the internal inequalities, since the city has a clear "oil stain" layout, with a consolidated and densely populated urban center and a polarized periphery [31].

The Catalan Urban Planning Act (revised text of 2010) establishes that the urban structure of the municipalities is made up of general and local systems, the facilities and a system of free public places. The system of free spaces includes parks, gardens, green areas and spaces for amusement, leisure and sport. The urban green system is usually formed in the urban fabric in a series of isolated elements that can have an important ecological value with respect to the continuity of habits. Therefore, the linear elements, such as walks, park ways or linear parks, behave similar to connectors, complemented by the tree-lined roads in the urban section. The interconnection between parks, gardens and interstitial spaces makes up a comprehensive green mosaic that increases biodiversity and implies an improvement in the quality of the public space.

The Tarragona Municipal Urban Planning Plan (2013) defines the municipality's system of free spaces in a broad sense. According to this document, this system includes the public parks and gardens, the ramblas (boulevards), squares and all the free, public green spaces located on urban, urbanizable or non-urbanizable land. Agenda 21 Local in Tarragona considers urban green to be the city spaces where natural elements penetrate in the form of parks or gardens, which are tree-lined, with water bodies or garden elements in the streets.

Between the years 2012 and 2017, the Environment Department of the Tarragona Town Hall quantified a total of 65 green areas intended for public spaces, totaling 371 Ha of urban green (3.71 km$^2$). These areas include the green spaces that are part of the urban section [32]. Out of this group of green spaces in the city of Tarragona, 14 were defined as urban parks [33]. These are distributed throughout the municipal territory, except in the residential estates in the east (Figure 1). These public facilities respond to very varied typologies, from landscaped urban squares, such as the Sant Antoni park, just over 1200 m$^2$

or 0.12 hectares, to extensive areas of natural vegetation, such as the fluvial part of the Francolí river, of 13.7 hectares.

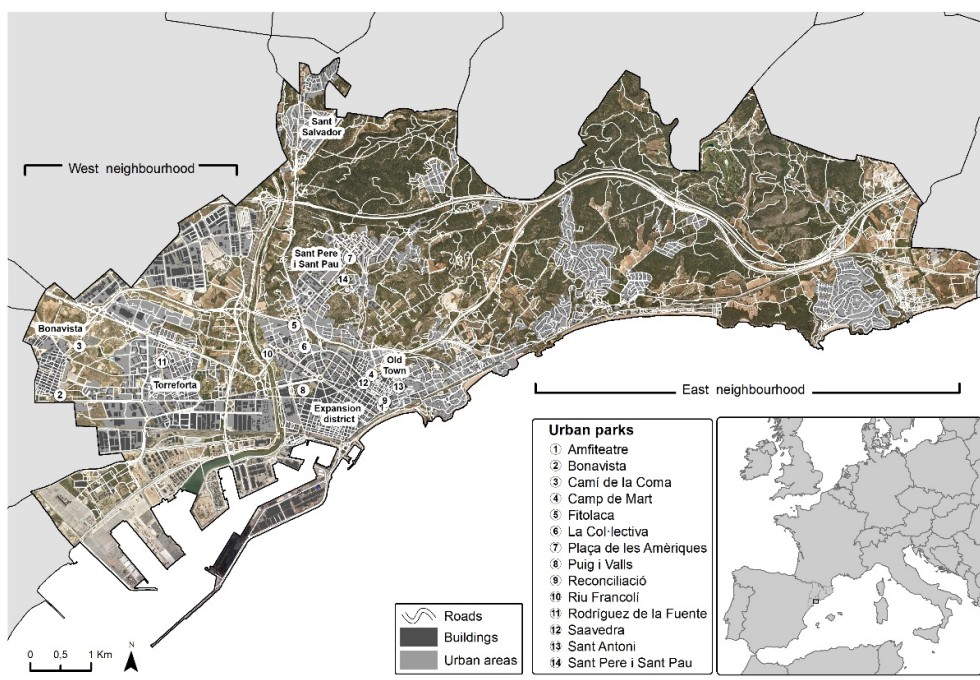

**Figure 1.** Location of the urban parks in Tarragona. Source: Own work. Orthophotomap base map 1:5000 of Cartographic and Geological Institute of Catalonia.

## 3. Methodology

The methodology applied in this work uses fieldwork, GIS and statistical analysis as shows Figure 2.

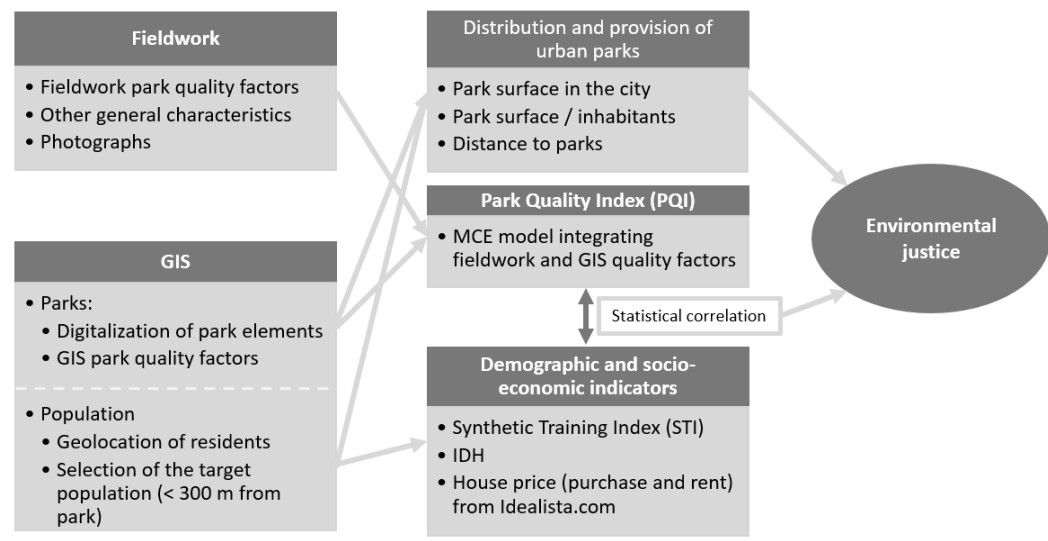

**Figure 2.** Methodological process.

Fieldwork permitted to obtain general information about parks, measure the quality factors of parks and take photographs. Use of a Geographical Information System (GIS) has made it possible to digitalize the elements that exist in the parks and their limits, geolocate the data regarding the population registered in the Municipal Register of Inhabitants, and select the target population; in other words, that lives within 300 m or less of an access

point of any of the parks and, finally, obtain part of the information needed to develop the MCE model's factors using spatial analysis operations and implementing them.

Finally, statistical analysis was carried out of the sample taken from the selection of inhabitants registered at a distance equivalent to or within 300 m of a park access point to characterize the socio-demographic perspective of the potential users of the said space.

With the set of indicators, both direct and indirect, that characterize the quality of the urban parks, a hierarchical and weighted Multi-Criteria Evaluation Model was designed, which has allowed us to obtain the PQI.

### 3.1. Obtaining General Information and Factors to Assess the Park Quality

The quality of urban parks is very important with respect to spatial and environmental justice [34]. The quality-of-life community factors for parks used in this work were selected from the Madrid case (Spain) [35] and Bucaramanga (Colombia) [36]. The model included a total of 20 factors. Some of them were obtained through cartographic analysis using GIS (ArcGIS 10.6.1), others were obtained from fieldwork and, most of them applying both methodologies (checking cartographic results with fieldwork) (Table 1).

**Table 1.** Factor name, description and methodology used to obtain them. Selection based on Canosa, Sáez, Sanabria and Zavala (2003) and Rivera (2015).

| Factors | Description | Method | |
|---|---|---|---|
| | | GIS | Fieldwork |
| Presence of vegetation | Percentage of area with vegetation in relation of the total area of the park without accounting for the shade they provide | X | |
| Green shade | Percentage of area with shade at noon in relation to the total area of the park | X | |
| Hazards or allergic vegetation | Number of plant species (dangerous or allergenic) | | X |
| Roads | Existence, adequacy and quality of roads and/or trails for the mobility of people | X | X |
| Architectural barriers | Presence of elements that hinder or prevent access and measures to overcome them | | X |
| Sports equipment | Spaces equipped to play football or basketball | X | X |
| Gym for seniors | Existence and status of gym for seniors | X | X |
| Enclosure fencing | Existence of separation of sports areas to avoid dangerous situations for other users | | X |
| Other sports equipment | Sports equipment for minority use such as hockey or volleyball courts | X | X |
| Facilities for cultural activities in open areas | Amphitheaters or bleachers | X | X |
| Facilities for cultural activities in closed areas | Indoor spaces suitable for cultural events | X | X |
| Other facilities | Kiosks or sales stands semi-permanent or seasonal | | X |
| Lighting | Presence of lighting with adequate power and distribution | X | X |
| Benches | Presence, proper distribution and status of benches | X | X |
| Other furnishings | Presence and condition of drinking water fountains, ponds or sculptures or monuments | X | X |
| Children's games | Presence and state of space with swings, slides, seesaws, etc. | X | X |
| Adult games | Presence and condition of petanque courts or areas set up for board games | X | X |
| Rubbish bins | Presence and status of rubbish bins | X | X |
| Dog potty area | Areas with specific sanitary treatments to prevent the transmission of infectious diseases from dogs | X | X |
| Toilets | Presence of toilets, state of conservation, operation and cleanliness | X | X |

Source: own work.

In each urban park in Tarragona, we generated a spatial and theme-based database with information on the urban location, the surface area covered by vegetation, the covered green shadow and the various facilities (Figure 3). This information was digitalized based on the Topographic Map 1:5000 and orthophotography on the same scale; both documents were provided by the Cartographic and Geological Institute of Catalonia (Institut Cartogràfic i Geològic de Catalunya).

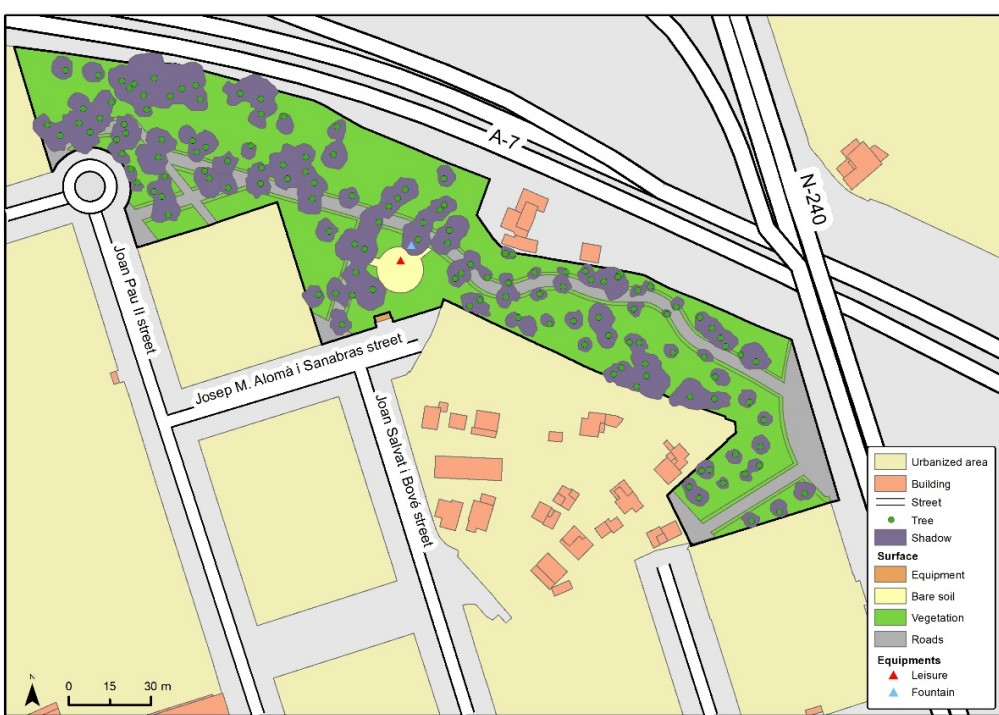

**Figure 3.** Example of the mapping database of the Fitolaca park (Tarragona). Source: own work.

Field work was also carried out, through the visits to the parks included in the study, between the spring and summer of 2018. The purpose of these visits was to obtain the direct information needed to establish the PQI (see Table 1), check the digitalized mapping and take photographs to characterize these spaces.

Each of these 20 factors was assessed on a scale of 0 to 3, where, in a standardized way, the value of 0 corresponds to the lowest quality of the factor, and the value of 3 refers to the highest quality of the factor.

### 3.2. Creating Demographic and Socio-Economic Population Indicators

Data from the Municipal Register of Inhabitants were used to characterize the population of the city of Tarragona on 31 December 2019. This database stores the residents' postal address and, therefore, it was possible to geolocate the registers based on this address and build up a mapping layer. The geolocation was carried out from the Instamaps platform of the Cartographic and Geological Institute of Catalonia (https://www.instamaps.cat/#/. Accessed from 1 December 2020 to 8 January 2021). The resulting layer was imported to ArcGIS and, using the tool "Near", inhabitants living within 300 m of the nearest park access point were selected and assigned to the nearest park, considering the mode of transport to be walking, because it is healthy and not affected by economic conditioning [37].

Some authors [38] use two factors to characterize the population demographics: the level of study and nationality. The level of study collected in the register of inhabitants in Tarragona refers only to the population aged 16 years old and over, and has been grouped into five categories (illiterate, no schooling completed, primary education, secondary education, university education). To compare the different territorial units, a Synthetic

Training Index (STI) was created based on the introduction of weighting for the population at each training level. The general formula of the STI is as follows:

$$STI = \frac{(\% \text{ illiterate} * 1) + (\% \begin{array}{c} \text{no schooling} \\ \text{completed} \end{array} * 2) + \left( \begin{array}{c} \% \text{ primary} \\ \text{education} \end{array} * 3 \right) + \left( \begin{array}{c} \% \text{ secondary} \\ \text{education} \end{array} * 4 \right) + \left( \begin{array}{c} \% \text{ university} \\ \text{education} \end{array} * 5 \right)}{5}$$

In this way, an index is obtained with values between 0 and 1, where 0 would be equivalent to the entire illiterate population and 1 would represent the opposite extreme, with the entire population having a university education. In order to neutralize the influence of the population's aged-based structure in terms of education (an older population tends to have lower education levels than a younger population), a direct standardization was carried out, based on the application of a typical population structure (the whole of the population of the city of Tarragona) to the 14 neighborhoods of the parks under analysis.

The second variable chosen is the origin of the population. In this case, we opted to use the population's place of birth, as opposed to nationality, because this addresses the idea of people from immigrant families who were born in Spain. To compare the different territorial units, the population born abroad was characterized using the average HDI value published by the United Nations Population Division in 2020. In addition, in order to better reflect the diversity of the population born in Spain, the HDI of the autonomous community of birth was taken into consideration. This information was obtained from the Valencian Institute of Economic Research (Instituto Valenciano de Investigaciones Económicas, year 2019).

In third place, due to the lack of disaggregated data at the income level, the population's economic characterization was indirectly analyzed based on housing prices. This information was taken from the property portal, Idealista.com, which allows you to consult the average renting and purchasing prices per square meter for apartments in a specific digitalized area. Thanks to this option, it was possible to define the 300 m area of influence around each park. The information obtained in this way is comparable with that from the Register of Inhabitants.

### 3.3. Creating the Park Quality Index (PQI)

The Multi-Criteria Evaluation (CME) encompasses a set of tools aiming to help decision-making [39], in which the various alternatives determined by multiple criteria and objectives are in conflict [40]. This work adopts the multi-criteria evaluation model in order to discover the degree of quality or suitability of urban parks, based on the selection of a series of indicators, subindicators and factors (Figure 4). To do this, the initial 20 factors (first hierarchical level) were grouped into seven subindicators (second hierarchical level) and, in turn, these were joined together in three indicators that correspond to (1) the quality of the vegetation, (2) the quality of the facilities and (3) the quality of the street furnishings (level 3). Finally, the combination of the three indicators leads to the Quality-of-Life Community indicator for parks (PQI) (level 4).

One of the essential characteristics of an MCE is the importance or weights according to the percentage of each factor, subindicator and indicator used in the model. The final result will largely depend on the weight that is assigned to each part of the model. In this case, the weight assignment is related to the established hierarchies and groups, so that they each add up to 100%. If we take the third hierarchical level as an example, and apply the decision formula or rule, vegetation is combined with a 40% weighting and facilities are combined with a 30% weighting, while the weighting for property is 30%. In order to perform the different aggregations of the model, we used the Weighted Overlay command in ArcGIS 10.6.1.

### 3.4. Statistical Analysis and Environmental Justice

Finally, once the IQP for each park was calculated, it was correlated with the demographic and socioeconomic characteristics of the population assigned to each park

(populations at a distance < 300 m). To measure the fit of the variables, the Pearson correlation index ($R^2$) was calculated using MS Excel software.

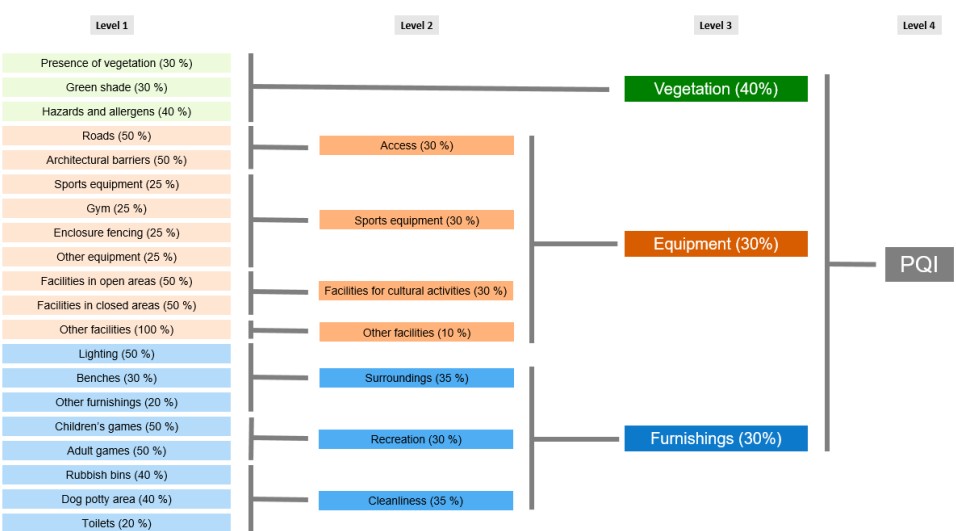

**Figure 4.** MCE of the quality of urban parks. Source: own work.

## 4. Results

The results of the research are structured in three areas. First, those related to the distribution and provision of the urban parks in the city; second, their quality based on calculation of the PQI; third, an attempt to determine whether environmental justice exists by relating, on the one hand, the quality of the parks and, on the other hand, the population's socio-demographic characteristics.

### 4.1. The Distribution and Provision of Urban Parks in the City

The distribution of parks in Tarragona reveals that they are concentrated in the central part of the city. This corresponds to the area with the highest density populations: out of the 14 parks, five are located in the Ensanche area (Saavedra, Camp de Mart, La Col·lectiva, Riu Francolí and Puig i Valls parks). This highlights the number of parks in the eastern area, with the Amfiteatre park, in Sant Antoni and the Reconciliació park. However, three aspects need to be specified: (a) this is the urban area with the largest surface area in the whole city; (b) the total surface area of the existing parks is not very big and most of them cover an average or small surface area; and (c) most are in the area nearest the city center, neighboring the historic center, where there is no urban park because of the morphological characteristics. The northern area has low-density residential estates and does not have these facilities (Figure 5). The existence of a high number of private urban green areas means that there are no public urban spaces. The districts to the west of the city have a high population and proportionally few urban parks.

According to the data taken from the land registry plots in each of the 14 urban parks listed by the town hall in the city of Tarragona, their total surface area is 374,606 sqm. However, this overall figure hides various case studies: the surface area differs considerably between the parks, with two of them, the Riu Francolí park (with 130,684 sqm.) and Sant Pere i Sant Pau park (with 122,130 sqm.), representing two thirds of the total surface area of urban parks (Figure 5). At the other extreme, we find parks that correspond more to the concept of landscaped square, such as the Sant Antoni square (1294 sqm.) and Fitolaca square (1558 sqm.).

There is also a difference between the occupancy percentages of each surface area type in the parks (green areas, bare soil, roads and facilities). Generally speaking, vegetation is the predominant type, as it covers a little more than half the area (55.1%), with much higher values in the cases of the Bonavista (81.5%) and, particularly, the Sant Pere i Sant Pau

(88.2%) parks. In the case of the latter, its large surface area makes it the city's "green lung". At the other extreme, we have the "landscaped squares" in Sant Antoni and Rodríguez de la Fuente, which have extremely low vegetation values (14.5% and 17.0%, respectively).

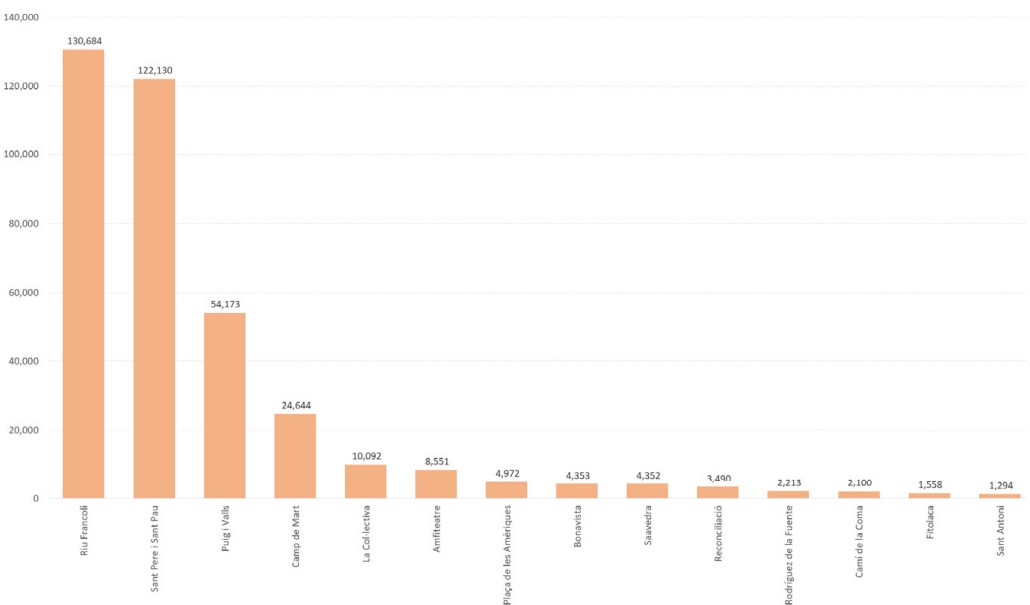

**Figure 5.** Surface area of the urban parks in Tarragona, in sqm. Source: own work.

Finally, it is worth highlighting the disparity between the surface areas occupied by the various facilities (sport, culture, etc.) in each park. In some parks, these elements are completely non-existent (La Col·lectiva and Rodríguez de la Fuente parks), while in others (Camp de Mart and Camí de la Coma park), at least one third of the surface area is covered by facilities.

*4.2. The Provision and Distribution of Urban Parks with Respect to the Population*

The surface area of the urban parks in relation to the inhabitants shows an average available surface area of 2.62 sqm. per inhabitant. This figure rises to 15.09 m$^2$/inhab. with all the green spaces (periurban) in the municipality. The value achieves the thresholds recommended by the World Health Organization, which establishes a figure of around 10–15 sqm./inhabitant. Regarding their accessibility, 39% of the population in the Tarragona capital live within 300 m of an urban park and the average distance in a straight line is 710 m, which is much higher than recommended (Table 2).

If the urban areas are used as the analysis unit, we can observe that the provision of urban park surface area per inhabitant is unequal according to the area of residence. This geographic distribution shows an important degree of environmental injustice. Therefore, some areas appear without any urban parks inside them (the historical center and Sant Salvador), whereby the rate is 0 sqm./inhabitat. In some areas, such as in the case of Bonavista, the parks lie in the periphery of the urban area. In others, such as Torreforta and Ensanche, there is a minimum provision per inhabitant, resulting from the combination of its relatively high population and the low surface area of the existing parks. The area with the residential estates in the eastern area (Levante) has below-average values, but, due to its urban characteristics, it is marked by a low density, the presence of private green areas and its disconnection from the city.

*4.3. The Quality of Urban Parks*

The average PQI in Tarragona is 53.39 points, with the highest assessment occurring in vegetation (67.86), followed by street furnishings (45.50) and facilities (32.32) (Table 3 and Figure 6). Out of the 14 urban parks analyzed, none of them showed a "good" quality (PQI equal to or higher than 70 points) and only four have a "medium-high" rating

(60–69 points). Specifically, these are the Puig i Valls park (62.81), the Riu Francolí park (62.22), the Amfiteatre park (61.42) and the Fitolaca park (60.29). Their overall score (total assessment of vegetation, facilities and furnishings) is due to different factors. Therefore, the Puig i Valls and Amfiteatre parks owe their high assessment to the vegetation (number of examples, green shadow and absence of allergenic species, 80.00 and 83.33, respectively), although they have lower scores for the other two indicators (facilities 39.00 and 32.08; furnishings 40.02 and 59.15, in the same previous order). On the contrary, the good score obtained by the Riu Francolí park is not due to the assessment of its vegetation (with 53.33 points, it is the third lowest urban green area in the city). Instead, it is due to the good quality of both its facilities (74.78) and its furnishings (67.43), in which it leads the city ranking as a whole.

**Table 2.** Average surface area available per inhabitant and average distance to the nearest urban park.

| Urban Parks | Total Population | | | Population Living < 300 m | | |
|---|---|---|---|---|---|---|
| | Inhabitants | Average Distance (m.) | sqm./Inhabitant | Inhabitants | Average Distance (m.) | sqm./Inhabitant |
| Amfiteatre | 4053 | 518.1 | 2.11 | 918 | 205.2 | 9.31 |
| Bonavista | 9945 | 337.1 | 0.44 | 4440 | 181.7 | 0.98 |
| Camí de la Coma | 2931 | 436.7 | 0.72 | 308 | 270.3 | 6.82 |
| Camp de Mart | 5618 | 294.3 | 4.39 | 2841 | 202.3 | 8.67 |
| Fitolaca | 1089 | 334.9 | 1.43 | 314 | 235.4 | 4.96 |
| Riu Francolí | 6799 | 207.3 | 1.48 | 6131 | 123.2 | 1.65 |
| La Col·lectiva | 6190 | 257.9 | 0.80 | 4374 | 189.4 | 1.14 |
| Pl. de les Amèriques | 20,867 | 1285.5 | 2.60 | 6296 | 170.6 | 8.60 |
| Puig i Valls | 32,105 | 369.3 | 0.11 | 13,046 | 154.1 | 0.27 |
| La Reconciliació | 2015 | 155.6 | 64.86 | 2015 | 155.6 | 64.86 |
| Rodríguez de la Fuente | 23,728 | 435.6 | 0.09 | 5331 | 172.7 | 0.42 |
| Saavedra | 9565 | 324.9 | 0.45 | 4245 | 196.0 | 1.03 |
| Sant Antoni | 15,168 | 2424.4 | 0.09 | 3004 | 157.2 | 0.43 |
| Sant Pere i Sant Pau | 2786 | 179.8 | 43.84 | 2395 | 137.2 | 50.99 |
| Total | 142,859 | 710.3 | 2.62 | 55,658 | 166.4 | 6.73 |

Source: own work.

**Table 3.** PQI values for each park and each indicator.

| Urban Parks | Vegetation (40%) | Facilities (30%) | Furnishings (30%) | PQI (100%) |
|---|---|---|---|---|
| Amfiteatre | 83.33 | 39.00 | 40.02 | 61.42 |
| Bonavista | 86.67 | 31.60 | 33.67 | 59.65 |
| Camí de la Coma | 36.67 | 54.28 | 52.00 | 44.90 |
| Camp de Mart | 63.33 | 35.48 | 41.34 | 50.87 |
| Fitolaca | 80.00 | 30.00 | 51.17 | 60.29 |
| Riu Francolí | 53.33 | 74.78 | 67.43 | 62.22 |
| La Col·lectiva | 80.00 | 15.65 | 47.00 | 55.66 |
| Pl. de les Amèriques | 46.67 | 26.00 | 62.83 | 45.54 |
| Puig i Valls | 80.00 | 32.08 | 59.15 | 62.81 |
| La Reconciliació | 63.33 | 14.40 | 45.58 | 46.66 |
| Rodríguez de la Fuente | 63.33 | 33.15 | 16.33 | 44.04 |
| Saavedra | 63.33 | 34.63 | 36.60 | 49.47 |
| Sant Antoni | 66.67 | 1.25 | 56.60 | 47.80 |
| Sant Pere i Sant Pau | 83.33 | 3.20 | 27.35 | 56.05 |
| Total | 67.86 | 32.32 | 45.50 | 53.39 |

Source: own work.

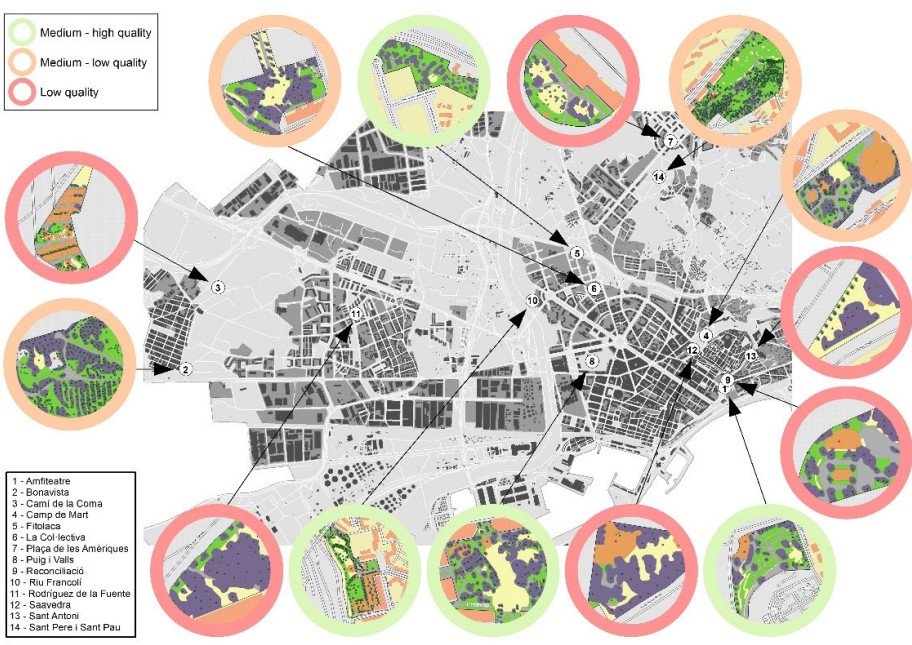

**Figure 6.** Location and quality level of the urban parks in Tarragona. Source: own work.

For the "medium-low" score (between the values 50 and 59 in the PQI), we find four parks: Bonavista, Sant Pere i Sant Pau, la Col·lectiva and el Camp de Mart. The good quality of their vegetation, which helps to offset their fairly mediocre scores in the other two PQI components, gives them an intermediary score. The most representative case in this respect is the Col·lectiva park, where the quality of the vegetation (80 points, the sixth highest) offsets the discrete score for its facilities (15.65).

Finally, we found six parks with a "low" assessment for their quality (PQI under 50 points): Saavedra, Sant Antoni, La Reconciliació, plaça de les Amèriques, Camí de la Coma and Rodríguez de la Fuente. In all of them, with the exception of the Camí de la Coma park, the best scoring indicator was for the vegetation, while, in the other two indicators (facilities and furnishings), low values were obtained (Table 3). The extreme cases are the furnishings score in the Rodríguez de la Fuente park (16.55) and, particularly, the score for the virtually non-existent facilities in the Sant Antoni park (1.25).

The results of the analysis reveal a meagre relationship between the location of the parks in the urban area and their quality. In terms of environmental justice, it could be expected that the parks located in the more peripheral areas were of a lower quality, while those in the center of the urban hub were better quality. However, in all the urban areas, there are parks with different assessments according to their PQI. To provide just one example in this case: two urban parks very close together, the Amfiteatre park and Reconciliació park, separated by just one street, have very different PQI values: while the first one has "medium-high" quality (61.42 points), the second one has "low" quality (46.66).

### 4.4. Environmental Justice Regarding Socio-Economic Characteristics: The Level of Studies, the Place of Birth and Housing Prices

As we can see from Figure 7a, the relationship between the PQI value and the standardized STI of the population living within 300 m reveals a degree of environmental inequity. On the one hand, with the positive value of the $R^2$ coefficient and, on the other hand, the actual positive slope of the trend line, it can be concluded that there is a causal relationship between the parks with a lower level of quality and the lower level of studies among the population living within 300 m. In spite of this, this relationship is not particularly robust, with an $R^2$ coefficient value of 0.1159. One factor that influences this behavior lies in the lower values of some urban parks, which are much lower than the other analyzed cases. In the urban parks with better values, the relationship between their

coefficient is not so clear; in other words, the higher values in a variable correspond to the highest scores in the other one.

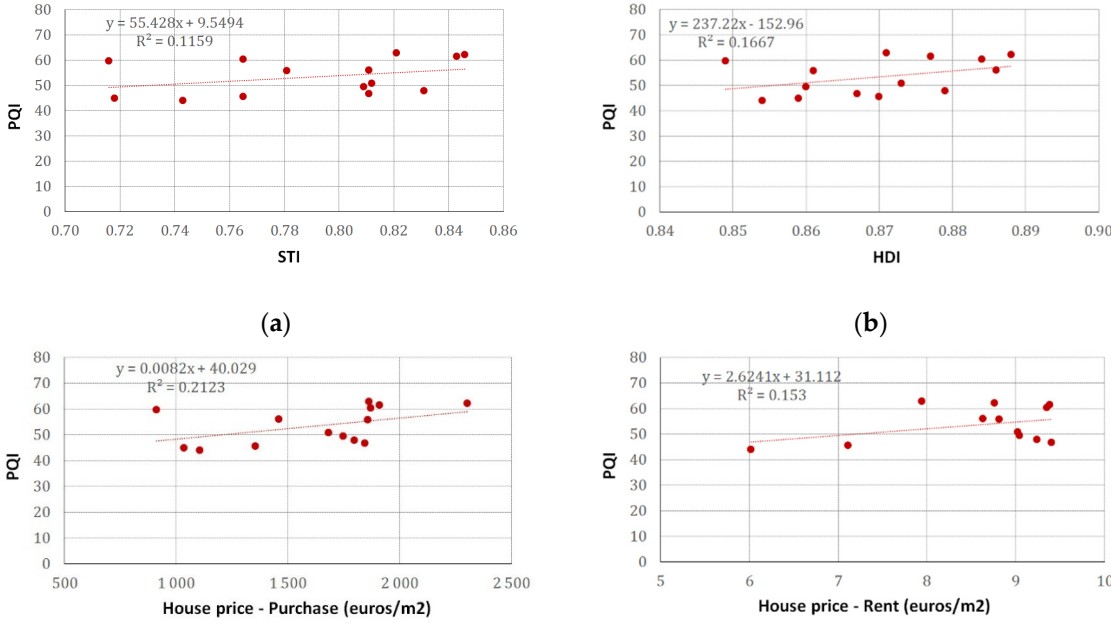

**Figure 7.** Correlation between the quality of the parks (PQI) and the socio-demographic variables of the population living within 300 m of the nearest park: (**a**) study level index for the population; (**b**) average value of HDI; (**c**) Average value of the purchasing price (euros/m$^2$) and (**d**) average value of the rental price (euros/m$^2$). Source: own work.

The results of the correlation between the HDI of the place of birth and the PQI reveal a clear correlation: the PQI value for each park and the average value of the HDI correlate with an R$^2$ of 0.1667 (Figure 7b).

The correlations between the PQI and the average price per square meter that is purchased or rented (Figure 7c,d, respectively) show fairly similar situations: if the dwelling is purchased or rented, its price is higher among those located near the greater quality parks. There is also a clearer relationship (R$^2$ = 0.2123) in the case of purchased dwellings, as the result of a direct linear correlation, whereby the lowest values of one variable are also related to the lowest indicators of the other variable and vice versa. It could be presumed that this direct relationship has something to do with the more or less central location of each park with respect to the city as a whole, understanding that the housing prices follow a more or less concentric logic, where the more central apartments have higher prices than those located in peripheral areas.

At any event, although this idea in the case of the dwellings is certain, it has already been mentioned that the quality of the parks does not follow this pattern, and so the robustness of the relationship is not due to this trend. A weaker, but equally positive (Figure 7c,d) relationship is the one between the price of home rentals and each park's PQI (R$^2$ = 0.1530).

## 5. Discussion

This article assesses the environmental justice in the city of Tarragona (Spain) with respect to the accessibility, availability and quality of the urban parks and the socio-demographic characteristics of its population. According to the Regional Office for Europe of the World Health Organization [22], the minimum surface area of urban green areas must be between 10 and 15 square meters per inhabitant, and within a distance of 300 m or 5 min walk from the dwelling. However, some authors [41] maintain that this strictly quantitative measurement of the provision of green spaces is not enough, and that the

parks must fulfil three basic conditions: availability (they are within a distance that allows its potential users to enjoy it), accessibility (when the user feels welcome, can access the park freely and use it for recreational purposes at any time) and, finally, a certain degree of attractiveness (when the space responds to individual needs, expectations and preferences).

A significant part of the work on environmental justice and urban parks has focused on assessing the distribution of green areas in the city [42,43] by calculating the distance between the place of residence and the nearest green area and using a geographic information system [44]. Even though the most frequently used unit is the distance in meters, some authors choose to measure accessibility using the travelling time according to the means of transport used (public transport, by foot, by bicycle and private car) [45]. Calculating these indicators using Euclidean and network distances shows a clear influence of the type of distance chosen (Euclidean versus the distance in the network). Therefore, we have to use these indicators carefully as planning support tools.

Analysis of accessibility, understood to be the physical or time separation between the actual location of the park and the users' place of residence, is complemented by architectural accessibility issues (for example, if it is a non-closed space); psychological accessibility (if it is attractive enough for potential users to visit) [41] or its "walkability", i.e., whether parks are accessible to people with limited mobility, such as children or the older population [46].

Some of the limitations found in this study are related to urban mobility. By selecting a population within a certain distance threshold with respect to a park, you start with the premise that the population only uses the urban parks in their own urban residency area, in other words, as if these urban divisions led to "islands" or self-contained compartments, without people moving from one to the other. Obviously, this is not entirely true, since mobility is a fundamental component of cities, conditioned by the place of residence and the locations that people visit regularly (work, leisure areas, daily shopping, children's study area). Therefore, beyond the analysis of the provision of urban parks based strictly on the place of residence, it would be important to note the population's daily mobility. The type of mobility used here is by foot, following the recommendations of the World Health Organization. However, due to the promotion of public transport as part of countries' commitment to reduce the consumption of hydrocarbon fuels, and the proliferation of Personal Mobility Vehicles (PMV), this segmented conception of the city has to change or, at least, reconsider the cut-off thresholds and account for other types of mobility. It is also necessary to mention that when selecting the population that are served by a park, the Euclidean distance, i.e., the distance in a straight line from the nearest park to the place of residence, was used, instead of the real distance using the city's road section. With this latter consideration, accuracy could be increased. On the other hand, the central location of the urban parks with respect to the city as a whole can influence the value of the homes located in more central areas, as opposed to others that are further away. In the case of the quality of the Tarragona parks, we did not find this association, and so the statistical correlation does not follow this trend.

The potential user public in each park (served population) is not made up of a uniform group of individuals with common demographic, economic and social characteristics. In fact, there is a consensus that social injustice regarding urban green areas usually comes along with a certain social stratification and/or residential segregation [47], which can lead to what some authors call "green gentrification" [43]. In this respect, some contributions examine the relationship between the number, proximity and quality of green areas, the socio-economic characteristics of the inhabitants [10,26,44,48] and the composition of the dwellings [48] In addition to these characteristics, consideration is often given to the ethical composition [28,29,49]. For example, De Suosa et al. (2018) [43], in their comparative work between Faro (Portugal) and Tartu (Estonia), observed significant inequalities in the housing districts of the socialist stage in the first of the cities, where most of the Russian minorities live (with a variable availability of public green spaces between 1.04 and 164.07 sqm. per inhabitant), whereas the Romanian communities in Faro were located

in districts without access to public green spaces, although there were smaller differences (from 1.22 to 31.44 sqm. per inhabitant). Other studies, on the other hand, focus on specific demographic groups such as young people [13,50] or immigrants [27]. The availability of this information and its high degree of territorial disaggregation in sources such as the Municipal Register of Inhabitants makes it easy to use, since it can be georeferenced. The data on the population's income are a different case, because, due to confidentiality issues, they are difficult to obtain. In this case, it is essential to use secondary sources to deduce the economic level of the population that a park serves.

The relationship between the disaggregated and precise socio-economic data made it possible to obtain positive correlations between these characteristics, park accessibility and quality, providing evidence of deficits in some areas in the city of Tarragona. Certain urban areas are under-endowed with urban parks vis-à-vis the land occupancy system and are mostly under dispersed forms and without general system reservations, beyond the needs arising from mobility. They require greater research attention. By comparing the different green areas with the neighborhood's social characteristics, it is possible to identify the priority areas and improve their condition, accessibility, quality and distribution [51]. This research has not segmented the socio-economic information on specific groups (according to age, origin), although it has standardized the values used for analyzing them. In future research, it would be relevant to consider the different user groups, the feeling of safety and security and social interaction.

This work has considered the quality of the parks as a factor of environmental justice because urban green areas are relevant for the urban quality of life and for promoting environmental equity [52]. Some authors [53] believe that in order to determine the environmental justice, accessibility and availability of green areas, you have to also consider their quality. The results of their analysis, combined with the socio-economic characteristics, broaden the understanding of environmental justice with respect to the parks. The literature on this issue has found numerous proofs of this. Corley et al. [30] established relationships between the various aspects involved in the quality of urban green areas, and Brown et al. [54] found significant associations between types of urban park and their benefits for the population. Another important conclusion drawn from these approximations is that improving and designing urban parks should consider resident preferences [55,56]. These actions would allow them to become community assets [57].

In response to this need, this work built the PQI: a synthetic index that can be used to assess environmental justice with greater precision than the availability of green surface areas or their distance. Another significant contribution from this work is that the lack of information on the population's income level was substituted by alternative sources. Therefore, to establish correlations with the PQI, the work instead used the level of studies, the resident population's HDI and home sale and rental prices within 300 m of a park. Out of these variables, the one that showed the strongest correlation was home sale prices.

Using the MCE techniques to build the PQI is an attempt to reduce subjectivism, but this always remains because choosing the factors and their weights is a subjective action. Differentiating between the selection of weights (vegetation 40%, facilities 30% and furnishings 30%) can provide a positive reading due to the possible actions aimed at increasing the quality of the parks: improvements to vegetation may require a greater economic effort which, in some cases, has a temporary repercussion (e.g., annual vegetation) or, in others, long-term results were obtained (e.g., time it takes for the tree vegetation to reach adult age). Intervention and maintenance regarding facilities and, particularly, furnishings, can lead to gains in the quality of the parks in a relatively economic way, in a short-term and with significant durability.

## 6. Conclusions

There is great disparity among the parks in the city of Tarragona: peripheral parks and central parks; large parks and landscaped squares. In terms of quality, vegetation is the parameter with the best score, while facilities and furnishings have deficiencies.

The value obtained for accessibility is lower than that recommended by the EU, and the same was true for the availability of green spaces, although this reduces if we consider green spaces overall. Based on the correlations between the PQI and the population's socio-economic characteristics, it can be stated that there is a causal relationship between these variables. However, the levels of environmental injustice are reduced. In this sense, in future work, the study could be improved by a more in-depth analysis of the type of vegetation, since certain types of vegetation provide different benefits to the population and, in some cases, can even cause harm; addition of other factors such as landscape quality assessment or new sources as the vegetation index of normalized difference (NDVI), which allows for an estimation of the quantity, quality and development of the vegetation; and, finally, incorporating the user's perception in the PQI model.

This work has shown that the relationship between access to green areas and environmental justice is complex. The aggregate results of the PQI as well as those of the sub-indicators and factors can be used by a competent administration to decide which parks should be a priority and which factors should be improved in each of them. The deficits in accessibility and quality can be overcome with strategies and actions, which will, on the one hand, increase the supply of green spaces in places with inequities and, on the other, increase the quality of these parks, with a greater endowment of cultural, recreational and sport facilities. The new urban agendas, especially in the post-pandemic context, include an analysis of urban habitability, reduction in social inequalities and improvements in health conditions; therefore, the planning, design and management of urban green areas should take socio-spatial attributes into account.

**Author Contributions:** Conceptualization, J.A., J.I.M.M. and Y.P.-A.; methodology, J.A. and Y.P.-A.; software, J.A., J.I.M.M., Y.P.-A. and E.B.P.; validation, J.A. and Y.P.-A.; formal analysis, J.A., J.I.M.M., Y.P.-A. and E.B.P.; investigation, J.A., J.I.M.M., Y.P.-A. and E.B.P.; resources, J.I.M.M.; data curation, J.A., Y.P.-A. and E.B.P.; writing—original draft preparation, J.A., J.I.M.M., Y.P.-A. and E.B.P.; writing—review and editing, J.A., J.I.M.M., Y.P.-A. and E.B.P.; visualization, J.A, and E.B.P.; supervision, Y.P.-A.; project administration, Y.P.-A.; funding acquisition, Y.P.-A. All authors have read and agreed to the published version of the manuscript.

**Funding:** This work is part of the research project entitled «El paisaje como valor colectivo. Análisis de su significado, usos y percepción social» (The landscape as a common value. Analysis of its significance, uses and social perception) (CHORA-CSO2017-82411-P), financed by the Ministry of Science, Innovation and Universities (National Programme for Fostering Excellence in Scientific and Technical Research 2018–2020), AEI/FEDER, EU and Department of Research and Universities of the Generalitat of Catalonia (2017SGR22).

**Institutional Review Board Statement:** Ethical review and approval were waived for this study, due to it did not collect critical personal information.

**Informed Consent Statement:** Not applicable.

**Data Availability Statement:** Raw data were generated at Tarragona City Council (Ajuntament de Tarragona). Derived data supporting the findings of this study are available within the article and from the corresponding author J.A. on request.

**Acknowledgments:** We acknowledge Lourdes Llorach de la Peña, Joan Jaume Iniesta Girona (Tarragona City Council/Ajuntament de Tarragona) and Robert Casadevall Camps and Marc Domínguez Mallafré (Universitat Rovira i Virgili) for their co-operation for documenting this work.

**Conflicts of Interest:** The authors declare no conflict of interest.

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
