# Peer review of "Environmental Justice and Urban Parks. A Case Study Applied to Tarragona (Spain)"

_urbansci, doi:10.3390/urbansci5030062_

Round 1

Reviewer 1 Report

Thank you for giving me the opportunity to review the manuscript entitled 'Environmental justice and urban parks. A case study applied to Tarragona (Spain)'. The topics covered are extremely topical and the need to increase knowledge about environmental justice and ecosystem services is growing. However, from my point of view, the manuscript has significant shortcomings in terms of both structure and content.

In general, it is necessary to structure almost the entire article differently in order to facilitate the understanding of the research carried out. There is a difficulty in reading and linking the various topics mentioned from section 1 to section 4. Section 5 will need to be revised based on the changes made in the previous sections. Some of the requested changes are very specific.

Lines 46-48: Cite at least two articles that support the claim you are making.

Lines 50-51: Cite at least two articles that support your claim. The following are given as examples:

- Battisti et al. 2019. Residential Greenery: State of the Art and Health-Related Ecosystem Services and Disservices in the City of Berlin. https://doi.org/10.3390/su11061815

- Säumel et al., 2021. The healthy green living room at one's doorstep? Use and perception of residential greenery in Berlin, Germany. https://doi.org/10.1016/j.ufug.2020.126949

Line 52 and line 57-61: The benefits that urban greenery provides to citizens are referred to as ecosystem services. Please refer to this terminology and cite some examples. We suggest reading the following article:

- Costanza et al., 2017. Twenty years of ecosystem services: How far have we come and how far do we still need to go? https://doi.org/10.1016/j.ecoser.2017.09.008

Line 57: Cite at least one article for item 6 that deals with similar issues. The following is given as an example:

- Larcher et al. 2021. Perceptions of Urban Green Areas during the Social Distancing Period for COVID-19 Containment in Italy. https://doi.org/10.3390/horticulturae7030055

Line 70: Write in full the meaning of the acronym PQI, and in the following lines also explain MCE and HDI, explaining better what they are.

Lines 85-86: Insert at least one citation to support this statement.

Section 2, i.e. lines 83-161 can be moved to the Discussion section, using only the parts that can be used to compare your results.

Section 4 is to be structured differently: If you have identified 3 well-defined stages, you should structure points 4.1 and following, as follows:

- 4.1 using a GIS and a MCE integrated into the former: it is necessary to explain in a much more in-depth way which software was used, to describe in depth the methodology applied, which are the bibliographical references you followed. In particular, the part relating to MCE should be described in greater depth, and how it relates to section 5 should be better explained, especially from 5.2 onwards.

- 4.2 field work: describe it in a little more detail.

- 4.3 statistical analysis: indicate which software, which analyses were carried out and why, and which references were used.

- There is no part that explains in detail what the PQI is and how it is calculated. How did you read and analyse the data/information you collected through the 3 stages mentioned above? In the text it is very difficult to understand this information. Moreover, it would be very difficult to replicate this method in other realities.

Are figures 4 and 5 really results of the application of your methodology (i.e. GIS) or are they simply data already available to the public administration and aggregated in a different way to explain the reality in which you are working? If the answer is the latter, please move this information to the section where you outline the study area.

Section 6 should be expanded with further comparisons between your study and similar studies related to the theme of environmental justice.

In the conclusion you state that in terms of quality, vegetation is the parameter with the best score. If possible, you could also argue that it is necessary to fully understand and distinguish the type of vegetation in future studies, as different vegetation provides different benefits to the population, and in some cases even provides disservices. Furthermore, how could the PQI be improved? Would satellite data, NDVI etc. be useful for its implementation? How could public administrations easily use your results?

Author Response

Dear reviewer, many thanks for your review and for your valuable comments. We considered them very carefully, and we attached our response to the system.

Reviewer 2 Report

The manuscript titled "Environmental justice and urban parks. A case study applied to 2 Tarragona (Spain)" intends to document the degree of environmental justice in the urban parks in the city of Tarragona by establishing a PQI and learning about the population’s socio-demographic characteristics. The empirical basis of the paper is a qualitative research study, including an MCE model which constructed with a GIS, which allow authors to establish the PQI. Also, include indirect, standardized indicators for the socio-economic characteristics of the population living within 300 meters of the park, such as the Synthetic Training Index (STI), the HDI or home sales prices in each sector and the PQI has been correlated with the population’s socio-economic characteristics to obtain the spatial justice results in terms of the availability and quality of the urban green areas. The analysis is divided into three well-defined stages grouping different tools and methodologies: (1) using a GIS and a MCE integrated into the former; (2) field work and (3) statistical analysis. Moreover, this article assesses the environmental justice in the city of Tarragona (Spain) with respect to the accessibility, availability and quality of the urban parks and the socio-demographic characteristics of its population.

The research is original; it could be characterized as novel and in my opinion important to the field, it also has the appropriate structure and language been used well. In the meanwhile, the manuscript has a nice extent (about 7,000 words), the tables (2) and figures (6) make the paper to reflect well to the reader. For this reason, paper has a "diversity look", not only tables, not only numbers, not only words.

The title is all right. I think the abstract reflects well the findings of this study; it has an appropriate length and described well the manuscript (209 with limit 200 words). The introduction is effective, clear, and well organized; it really introduced and put into perspective what research is negotiating.

For the Methodology chapter, the research conduct has been tested in several areas of the world, with similar results and will probably be tested in others. In this way it is documented that a project which is tested in a place with its own characteristics can be implemented in other places around the world. Please revise the references of the manuscript and include references which are already exists in bibliography. I would be much more satisfied if the number of references was slightly higher (about 10 - 15 references) and I would appreciate it if also included data from the entire world (Asia, America, Europe and Australia e.tc.). In this way it is documented that a project which is tested in a place with its own characteristics can be implemented in other places around the world. Moreover, I would appreciate it if also saying more about the PQI (a paragraph or two I think it will be all right).

The results and discussion sections are very good. The argument flows and is reinforced through the justification of the way elements are interpreted. The same applies to the conclusions, which it could be longer.

It is advised to revise the Discussion and Conclusion. Both sections should be consistent in terms of Proposal, Problem statement, Results, and of course, future work. Your conclusion section is too short and does not do justice to your work. Make it your key contributions, arguments, and findings clearer. You must refer to the literature and previous studies in your discussion and conclusion sections. It is recommended to remove paragraphs from Results and Discussion and put them in the Conclusion, with nice order and to be enriched.

Please, revise the references, they must have an appropriate style, for this reason I would be good to reduce [see: Instructions for Authors / Manuscript Preparation / Back Matter / References: - (https://www.mdpi.com/journal/urbansci/instructions or https://www.mdpi.com/authors/references)].

Please, revise the reference “14. World Health Organization. Regional Office for Europe. Urban green spaces and Health. A review of evidence. Copenhagen, 2017”, line 566 and type the year correctly. I think must be revised as “14. World Health Organization. Regional Office for Europe. Urban green spaces and Health. A review of evidence. Copenhagen, 2016” or if you use the newer version, as “14. World Health Organization. Regional Office for Europe. Urban Green Spaces Interventions and Health. A review of impacts and effectiveness. Copenhagen, 2017”.

Please, revise the reference “20. Rigolon, A.; Toker, Z.; Gasparian, N. Who has more walkable routes to parks? An environmental justice study of Safe Routes to Parks in neighborhoods of Los Angeles. Journal of Urban Affairs 2017, 40 (4), pp. 1-16”, lines 577 – 578 and type the page correctly and use the appropriate style. I think must be revised as “Rigolon, A.; Toker, Z.; Gasparian, N. Who has more walkable routes to parks? An environmental justice study of Safe Routes to Parks in neighborhoods of Los Angeles. J. Urban Aff. 2018, 40, 576–591, doi:10.1080/07352166.2017.1360740”.

Please, revise the references 16 & 22 are the same, lines 567 – 568 & 581 – 582 and use the appropriate style. I think must be revised as “De Sousa Silva, C.; Viegas, I.; Panagopoulos, Τ.; Bell, S. Environmental Justice in Accessibility to Green Infrastructure in Two European Cities. L.  2018, 7, doi:10.3390/land7040134”. Do not forget the numbering of references it will be change.

Please, revise the reference “50. Fors, H; Molin, J. F.; Murphy, M. A.; Bosch, C. K. van den. Use participation in urban green spaces – For the people or the parks? Urban Forestry & Urban Greening 2015, 14, pp. 722-734”, lines 636 – 637 and type the title correctly and use the appropriate style. I think must be revised as “50.Fors, H.; Molin, J.F.; Murphy, M.A.; Konijnendijk van den Bosch, C. User participation in urban green spaces – For the people or the parks? Urban For. Urban Green. 2015, 14, 722–734, doi:https://doi.org/10.1016/j.ufug.2015.05.007”.

Please, revise the line 263 “Instituto Valenciano de Investigaciones Económicas” and place the institution in English “The Valencian Institute of Economic Research”, the same in the other lines of manuscript. I know that it is difficult to change the parks, which you can keep it. See line 228

Please, revise the line 335 “2.62 sqm. per inhabitant. This figure rises to 15.09 m2/inhab.” and place everywhere in the manuscript the same form “m2/inhabitant” if you agree. See lines 118, 120, 338, 350, 440 and Table 1. Moreover, revise the lines 164 and 166 and replace square kilometers with km2.

Please, revise the line 441 and replace “metros” with “meters”. 

Author Response

Dear reviewer, many thanks for your review and for your valuable comments. We considered them very carefully and we attached the response letter to the system.

Round 2

Reviewer 1 Report

After making several important changes to the text, the manuscript can be published.